# Involvement of the Microglial Aryl Hydrocarbon Receptor in Neuroinflammation and Vasogenic Edema after Ischemic Stroke

**DOI:** 10.3390/cells10040718

**Published:** 2021-03-24

**Authors:** Miki Tanaka, Masaho Fujikawa, Ami Oguro, Kouichi Itoh, Christoph F. A. Vogel, Yasuhiro Ishihara

**Affiliations:** 1Program of Biomedical Science, Graduate School of Integrated Sciences for Life, Hiroshima University, Hiroshima 739-8521, Japan; mikit-313@hiroshima-u.ac.jp (M.T.); masahof05@gmail.com (M.F.); aoguro@hiroshima-u.ac.jp (A.O.); 2Laboratory for Pharmacotherapy and Experimental Neurology, Kagawa School of Pharmaceutical Sciences, Tokushima Bunri University, Kagawa 769-2193, Japan; itoh@kph.bunri-u.ac.jp; 3Department of Environmental Toxicology, University of California, Davis, CA 95616, USA; cfvogel@ucdavis.edu; 4Center for Health and the Environment, University of California, Davis, CA 95616, USA

**Keywords:** ischemia, edema, AhR, inflammation, p47phox

## Abstract

Microglia are activated after ischemic stroke and induce neuroinflammation. The expression of the aryl hydrocarbon receptor (AhR) has recently been reported to elicit cytokine expression. We previously reported that microglial activation mediates ischemic edema progression. Thus, the purpose of this study was to examine the role of AhR in inflammation and edema after ischemia using a mouse middle cerebral artery occlusion (MCAO) model. MCAO upregulated AhR expression in microglia during ischemia. MCAO increased the expression of tumor necrosis factor α (TNFα) and then induced edema progression, and worsened the modified neurological severity scores, with these being suppressed by administration of an AhR antagonist, CH223191. In THP-1 macrophages, the NADPH oxidase (NOX) subunit p47phox was significantly increased by AhR ligands, especially under inflammatory conditions. Suppression of NOX activity by apocynin or elimination of superoxide by superoxide dismutase decreased TNFα expression, which was induced by the AhR ligand. AhR ligands also elicited p47phox expression in mouse primary microglia. Thus, p47phox may be important in oxidative stress and subsequent inflammation. In MCAO model mice, P47phox expression was upregulated in microglia by ischemia. Lipid peroxidation induced by MCAO was suppressed by CH223191. Taken together, these findings suggest that AhR in the microglia is involved in neuroinflammation and subsequent edema, after MCAO via p47phox expression upregulation and oxidative stress.

## 1. Introduction

The aryl hydrocarbon receptor (AhR) belongs to the superfamily of basic helix-loop-helix/Per-AhR nuclear translocator (ARNT)-Sim (bHLH/PAS) domain-containing proteins and acts as a transcription factor. Exogenous environmental chemicals such as 2,3,7,8-tetrachlorodibenzo-p-dioxin (TCDD) and benzo(a)pyrene (BaP), which bind to AhR to induce its transport from the cytosol to the nucleus, are classical AhR ligands. In the nucleus, AhR forms a heterodimer with ARNT and binds to dioxin/xenobiotic response elements (DREs/XREs) in the promoter regions of ligand-metabolizing enzymes, such as CYP1A1 and UGT1A1. Thus, the AhR pathway is an endogenous detoxification mechanism against harmful exogenous compounds [1].

It has been revealed that AhR is closely related to inflammatory reactions. Many inflammatory molecules, such as cytokines and chemokines, are induced downstream of AhR activation. Induction of tumor necrosis factor alpha (TNFα) by lipopolysaccharide (LPS) in the microglia has been reported to be suppressed by siRNAs targeting AhR and in AhR knockout mice [2]. We previously reported that the interleukin-1β (IL-1β) gene has a classical DRE in its promoter region and is regulated by TCDD via protein kinase A and the CCAAT-enhancer-binding protein β pathway [3]. Interestingly, NF-κB, a master transcription factor of inflammation, can induce AhR activation to enhance the transcription of several inflammatory molecules [4]. In addition, indoleamine 2,3-dioxygenase (IDO), which metabolizes tryptophan to kynurenine (KYN), is reportedly induced by AhR [5], and importantly, KYN is an endogenous agonist of AhR. Together, these findings suggest that AhR signaling could form a positive feedback loop to potentiate inflammation.

Recently, AhR was reported to exacerbate infarct formation induced by ischemic stroke [6]. Ischemic conditions lead to the upregulation of tryptophan 2,3-dioxygenase expression but not IDO expression to increase the synthesis of KYN, which can strongly activate AhR. Activated AhR inhibits cAMP response element-binding protein signaling (CREB), and thus, suppresses survival pathways. In particular, decreases in brain-derived neurotrophic factor levels might be important for ischemia-induced neuronal injury. Kwon et al. demonstrated that AhR is also involved in cerebral ischemia–reperfusion injury [7]. Treatment with trimethoxyflavone suppresses infarct and edema formation induced by ischemia–reperfusion, although the mechanism is unclear. Collectively, these results indicate that, in the brain, AhR can damage neuronal cells during ischemia via multiple pathways.

We previously reported a method for visualizing edema by combining 2,3,5-triphenyltetrazolium chloride (TTC) staining and magnetic resonance imaging (MRI) and revealed that regulation of the progression of ischemic edema by activated microglia, brain immune cells, is critical for infarct progression and neuronal dysfunction [8]. Proinflammatory cytokines and reactive oxygen species (ROS) are thought to be important in the pathophysiology of ischemic stroke. ROS can activate NF-κB to upregulate proinflammatory cytokine expression [9]. Cytokines increase blood–brain barrier (BBB) permeability by decreasing the expression of tight junction proteins, claudins and occludins [10,11,12]. Based on these findings, in this study we examined the role of AhR in ischemic neuronal injury, focusing on neuroinflammation and edema.

## 2. Materials and Methods

### 2.1. Animals

All animal procedures were performed in accordance with the Fundamental Guidelines for Proper Conduct of Animal Experiments and Related Activities in Academic Research Institutions under the jurisdiction of the Ministry of Education, Culture, Sports, Science and Technology, Japan. The Animal Care and Use Committee of Hiroshima University approved the experimental protocols (No. C18-16-4). Male ICR mice and pregnant ICR mice were obtained from Japan SLC (Shizuoka, Japan) and were maintained in a temperature-controlled animal facility in a 12-h light–dark cycle.

### 2.2. Permanent Middle Cerebral Artery Occlusion (MCAO)

Permanent middle cerebral artery occlusion (MCAO) was induced according to our previous report [8]. Mice were anesthetized with isoflurane (Escain, MERCK, Kenilworth, NJ, USA). A midline incision was made in the neck to allow access to the left carotid bifurcation and the external and internal carotid arteries. After ligation of the vessels, a small incision was made in the external carotid artery, and a round-tip monofilament (filament size of 6–0) was inserted and gently advanced through the internal carotid artery until the tip occluded the origin of the middle cerebral artery. The monofilament was secured in place with a ligature, and the skin incision was closed with surgical clips.

### 2.3. Determination of the Infarct Size

The infarct size was assessed by TTC staining. Brains were sliced into 1-mm-thick coronal sections based at 0.0 mm from bregma and stained with 1% TTC solution in PBS at 37 °C for 10 min. TTC-stained brain sections were analyzed by ImageJ software (National Institutes of Health, Bethesda, MD, USA).

### 2.4. MRI

MRI was performed according to our previous report [13]. Briefly, mice were anesthetized with isoflurane, and body temperature was maintained at a constant 37 ± 0.2 °C. MRI data were acquired using an MRmini-SA system (DS Pharma Biomedical, Osaka, Japan) consisting of a 1.5-Tesla permanent magnet, a compact computer-controlled console, and a solenoid MRI coil with a 30 mm inner diameter. T2-weighted images (T2WIs) were obtained with the following parameters: repetition time (ms)/ echo time (ms) = 2500/69 and number of excitations = 4. To measure the signal intensity on T2WIs, the mean signal intensity was determined using INTAGE Realia Professional software (Cybernet Systems Co. Ltd., Tokyo, Japan) and ImageJ software.

### 2.5. Brain Water Content

The brain was rapidly removed and dissected into two regions, the cerebral cortex and the striatum. The wet weight of each brain region was measured, and the samples were then placed in an oven (100 °C) for 24 h and reweighed (dry weight). The percentage of brain water content was calculated as [(wet weight−dry weight)/wet weight of brain tissue] × 100 (%).

### 2.6. Modified Neurological Severity Scores (mNSSs)

The mNSS is a composite score of motor, sensory, balance and reflex tests on a scale of 0 to 18 (normal score = 0; maximal deficit score = 18). Scoring was performed according to a previous report [14].

### 2.7. THP-1 Cell Culture and Differentiation

The human monocytic cell line THP-1 was obtained from ATCC (TIB-202). THP-1 cells were differentiated according to our previous report [15]. Briefly, the cells were cultured for 2 days in the presence of 160 nM phorbol 12-myristate 13-acetate and then incubated for an additional 2 days with fresh media.

### 2.8. Total RNA Extraction and Real-Time PCR

mRNA levels were determined according to the protocol described in our previous report [16]. The primer sequences are presented in Table 1. mRNA levels were normalized to the level of the housekeeping gene *β-actin*, and levels in the treated samples were divided by those in the untreated samples to calculate the relative mRNA levels.

### 2.9. Immunoblotting

Immunoblotting was performed as described previously [16]. Briefly, cells were collected and lysed with radioimmunoprecipitation assay buffer (25 mM Tris-HCl (pH 7.6), 150 mM NaCl, 1% Nonidet P-40, 1% sodium deoxycholate, and 0.1% SDS). Equal amounts of protein were loaded, separated via SDS-PAGE and transferred onto polyvinylidene difluoride membranes. Blocked membranes were incubated with the following primary antibodies: anti-p47phox (610354, BD Biosciences, San Jose, CA, USA), anti-NADPH oxidase (NOX)-2 (sc-130543, Santa Cruz, Dallas, TX, USA) and anti-α-tubulin (T5168, Sigma-Aldrich). Then, the membranes were incubated with peroxide-conjugated secondary antibodies (Thermo Fisher Scientific, Waltham, MA, USA) and visualized using peroxide substrates (SuperSignal West Pico, Thermo Fisher Scientific).

### 2.10. Measurement of NOX Activity

NOX activity was evaluated by determining the levels of superoxide anion radials generated mainly by NOX as described in our previous report with slight modification [17].

### 2.11. Promoter Analysis

Promoter analysis of the 5′-upstream regulatory region of human and mouse IL-33 was performed with the TFSEARCH program [15,18]. Genetic sequence data from GenBank were used for the analysis.

### 2.12. Chromatin Immunoprecipitation (ChIP) Assay

Cells were fixed in 1% formaldehyde for 10 min at room temperature, and immunoprecipitation was performed with an anti-AhR antibody (ab2769, Abcam, Cambridge, UK) or control IgG antibody with the Pierce Agarose ChIP Kit (Thermo Fisher Scientific) according to the manufacturer’s instructions. The resulting immunoprecipitates, which included DNA, were analyzed by real-time PCR using Fast SYBR Green Master Mix (Applied Biosystems). The primers used are listed in Table 2. Human genomic DNA extracted from THP-1 macrophages was used as a positive control (Cp values were around 11). Amplification of target sequences was confirmed by the DNA sequence. The percentage input of each sample was calculated from the Ct values.

### 2.13. Determination of Lipid Peroxide Levels

The content of thiobarbituric acid-reactive substance (TBARS), which was used as an index of lipid peroxidation, was estimated as described in our previous report [19]. Briefly, the cerebral cortex was homogenized in 1.15% KCl solution. The homogenate was mixed with 7 mM sodium dodecyl sulfate, 16 mM thiobarbituric acid and 340 μM dibutylhydroxytoluene in acetic acid at a pH of 3.5. The mixture was incubated at 100 °C for 60 min, and then extraction was performed with 1-butanol-pyridine (15:1) solution. Then, the absorbance of this extract at 532 nm was measured. 1,1,3,3-Tetraethoxypropane was used as the standard.

### 2.14. Isolation and Culture of Mouse Primary Microglia

Cultures of primary microglia were prepared from 0- to 1-day-old ICR mice, according to the previous reports with some modification [9,20]. The forebrain was dissociated and the cells were plated in a poly-l-lysine-coated plastic culture flask with tissue culture medium, which consisted of Dulbecco’s modified Eagle’s medium (DMEM) supplemented with 10% fetal bovine serum (FBS) and 5 μg/mL insulin. After 7 to 10 days culture, cell dissociation solution including 0.25 U/mL collagenase D (Roche Diagnostics KK, Tokyo, Japan), 8.5 U/mL dispase II (Roche Diagnostics KK), 0.25 U/mL DNaseI (Sigma) and 0.1 μg/mL tosyl-L-lysyl-chloromethane hydrochloride) was added into the flask. The flasks were agitated on a shaker for 20 min and then the culture medium was added to stop the digestion reaction. After gentle triturating, cells were passed through a 70 μm cell strainer to obtain a single cell suspension. Cells were centrifuged at 200× *g* for 6 min and then the resulting pellets were resuspended in PBS including 5% FBS and 1 mM EDTA.

Magnetic separation of microglia was performed using the EasySep mouse CD11b Positive Selection Kit II (STEMCELL Technologies, Veritas Corporation, Tokyo, Japan), according to the manufactures’ instruction. Briefly, the cells were mixed with component A and B mixture and then were incubated for 5 min at room temperature. RapidSpheres (magnetic particles) were added into the mixture and incubated for 5 min at room temperature. Labeled microglia were isolated using EasySep Magnet and seeded at a density of 1 × 10^6^ cells/mL. After culture for 24 h, cells were used for each experiment. The purity was over 98% confirmed by Iba1 staining [21].

### 2.15. Statistical Analyses

The data are expressed as the mean ± S.E. Statistical analyses were performed using one-way analysis of variance (ANOVA) followed by Student’s *t*-test or Dunnett’s test. *p* values < 0.05 were considered statistically significant.

## 3. Results

### 3.1. Brain Injury Induced by Permanent MCAO Is Alleviated by AhR Antagonism

In mice, MCAO induces ischemia mainly in the striatum and cerebral cortex [22]. The ischemic region was confirmed by MRI using 3-hydroxymethyl-proxyl as a probe (data not shown). We first investigated AhR mRNA expression in the ischemic brain. AhR mRNA levels were significantly increased in both the cerebral cortex and striatum 3 h after MCAO, and this increase was maintained at 6 h (Figure 1). Minocycline is a strong inhibitor of microglial activation and inhibits nuclear translocation as well as mRNA expression of NF-κB to suppress proinflammatory cytokine induction [23,24]. Pretreatment with minocycline clearly suppressed AhR upregulation at all time points (Figure 1), suggesting that microglial AhR expression is elevated during ischemia. CYP1A1 is one of main targets of AhR [25]. When CYP1A1 expression was measured during ischemia, CYP1A1 mRNA levels increased in both the cerebral cortex and striatum at 3 and 6 h (Figure 2). Pretreatment with an AhR antagonist, CH223191, significantly suppressed CYP1A1 expression (Figure 2). Therefore, microglial AhR is considered to be functional during ischemia.

Growing evidence shows that AhR moderates proinflammatory cytokine expression and thus can control inflammatory reactions. Thus, we assessed the expression of TNFα, which is a proinflammatory cytokine regulated by AhR [21], during ischemia. MCAO induced expression of TNFα in the cortex and striatum in a time-dependent manner, and pretreatment with CH223191 significantly abolished TNFα expression (Figure 3A). In addition, the expression levels of IL-1β and cyclooxygenase-2 (COX-2) were also elevated 3 h after MCAO, and administration of CH223191 clearly suppressed the increment (Figure 3B), showing that AhR-mediated inflammation occurs during ischemia.

We previously reported that microglia activated by MCAO mediate vasogenic edema progression and induce infarct formation [8]. Thus, we next investigated the edema and infarct areas by MRI and TTC staining. T2WIs can be used to visualize both the infarct and edema regions, and TTC staining can identify the infarct. Thus, the size of the edema region can be determined by calculating the difference in the size of T2WI hyperintense region from that of the TTC-unstained area. When T2WIs were merged with TTC-stained images 24 h after MCAO, a T2 hyperintense region was observed around the infarct areas (Figure 4A,B), indicating that vasogenic edema surrounded the infarct area. We previously reported that vasogenic edema occurred at the striatum 6 h after ischemia and then spread into the cerebral cortex [8]. The vasogenic edema region was localized in the striatum and rarely spread into the cerebral cortex following pretreatment with CH223191 (Figure 4A,B). The infarct volume was reduced by CH223191 treatment (Figure 4A,B). The brain water content was increased by MCAO and was restored to normal levels by treatment with CH223191 (Figure 4C). mNSSs were largely worsened by ischemia compared with sham-operated mice (mNSS score = 0) and were improved by administration of CH223191 (Figure 4D). Taken together, these findings suggest that microglial AhR could have a role in eliciting neuroinflammation and promoting the progression of vasogenic edema, which can induce neuronal injury.

### 3.2. The NOX Subunit p47phox Is a Target of AhR in Inducing Oxidative Stress and Inflammatory Reactions

NOX is a superoxide radical-producing enzyme that has an important role in host defense [26]. NOX consists of several subunits, and the p47phox subunit is a regulator of the phagocytic NOX subunit NOX2, which generates superoxide anion radicals. It has been reported that p47phox has a DRE in its promoter region and thus can be regulated by AhR [27]. Therefore, we next examined the role of NOX in neuroinflammation.

THP-1 macrophages were used to determine whether AhR is involved in p47phox expression and function because we previously reported that THP-1 cells express functional AhR [15]. When THP-1 macrophages were treated with the AhR ligand TCDD or KYN, p47phox mRNA expression was significantly increased, but the expression of other subunits, namely, NOX2, p22phox, p40phox, p67phox and Rac-1, were not affected by AhR ligands (Figure 5A). p47phox expression was also induced under LPS-mediated inflammatory conditions in THP-1 macrophages (Figure 5A). The protein levels of p47phox were increased by stimulation with TCDD or KYN, and this increase was enhanced under LPS-stimulated conditions (Figure 5B and Appendix A). Pretreatment with CH223191 significantly suppressed p47phox expression induced by TCDD, KYN and the combination of LPS and TCDD or LPS and KYN (Figure 5C), indicating that p47phox is markedly induced under inflammatory conditions in an AhR-dependent manner.

We found two DREs in the promoter region of p47phox; one was previously reported (DRE2 in Figure 6A), but the other was novel (DRE1 in Figure 6A). ChIP clearly showed that AhR activated by TCDD or KYN was recruited to both DRE1 and DRE2 in the promoter region of p47phox (Figure 6B). The superoxide-generating activity of NOX was increased by treatment with TCDD or KYN, and this activity was significantly enhanced under LPS-induced inflammatory conditions in THP-1 macrophages (Figure 7). TNFα expression was increased by treatment with TCDD and KYN, and this increase was clearly suppressed by pretreatment with superoxide dismutase (SOD) or the NOX inhibitor apocynin (Figure 8). Treatment with SOD or apocynin alone did not affect TNFα expression (data not shown). Therefore, ROS generated by NOX is thought to be involved in TNFα expression. Taken together, these findings suggest that AhR stimulation can increase p47phox expression, which is responsible for TNFα expression via superoxide production, especially under inflammatory conditions.

Next, we examined the effects of AhR agonists on microglial p47phox. BV-2 microglia cell line did not express AhR under inflammatory condition (data not shown) and thus, we challenged mouse primary microglia. Treatment with TCDD as well as KYN significantly increased p47phox expression in mouse primary microglia (Figure 9). Taken together, AhR-dependent p47phox upregulation might be involved in oxidative stress and neuroinflammation.

### 3.3. MCAO Induces p47phox Upregulation and Subsequent Oxidative Brain Injury

The expression of p47phox mRNA was increased 3 and 6 h after MCAO in both the striatum and cortex (Figure 10A). Pretreatment with minocycline significantly suppressed p47phox expression induced by MCAO (Figure 10A), suggesting that p47phox expression is upregulated through activation of microglia during ischemia. TBARS levels were significantly elevated 24 h after MCAO in the striatum and cortex (Figure 10B). Administration of CH223191 clearly suppressed the increase in TBARS levels in the cortex (Figure 10B). Therefore, AhR-dependent oxidative stress can be elicited during ischemia.

Tryptophan 2,3-dioxygenase and IDO are rate-limiting enzymes in the tryptophan metabolizing pathway [28]. Several tryptophan metabolites, such as KYN, show agonistic activity for AhR. IDO expression was enhanced 3 and 6 h after MCAO (Figure 11), suggesting KYN as an AhR activator during ischemia. Further study is needed to show a mechanism of AhR activation during ischemia.

## 4. Discussion

The involvement of AhR in brain ischemic injury was first reported by Cuartero et al. [6]. Neuronal AhR activation after cerebral ischemia inhibits CREB signaling and pro-survival pathways subsequently activated by CREB, such as increased expression of brain-derived neurotrophic factor and bcl-x. Chen et al. reported that neural AhR activation following acute ischemic stroke increases astrogliosis and suppresses neurogenesis [29]. AhR is also reportedly expressed in microglia and can contribute to neuroinflammation [2,30]. We showed in this study that microglial AhR expression was upregulated during ischemia and that microglial AhR could be involved in oxidative stress and subsequent inflammatory reactions, suggesting that AhR plays a fundamental role in the pathological process of ischemia. The expression of the NOX subunit p47phox, which can stimulate the production of superoxide anion radicals followed by proinflammatory cytokine expression, was upregulated in microglia during ischemia. Thus, both neuronal AhR and microglial AhR are thought to aggravate ischemic neuronal injury but do so through different mechanisms of action.

AhR expression in microglia was elevated 3 h after ischemia. We previously reported that NF-κB RelA is a critical component regulating the expression of AhR and the induction of AhR-dependent gene expression [4]. During ischemia, damage-associated molecular patterns (DAMPs) are released from dying cells, and immune cells are sequentially activated by DAMPs and then induce inflammatory reactions [31]. In this study, TNFα expression was upregulated 3 h after MCAO. Additionally, the expression of other inflammatory molecules, namely, IL-1β and COX-2, was increased 3 h after MCAO, indicating that inflammation occurs at an early time point after ischemia. Therefore, inflammation accompanied by ischemia can increase AhR expression. AhR is known to increase the expression of proinflammatory molecules [32], and thus, AhR expression upregulation during ischemia might be one of the mechanisms by which the inflammatory reaction is amplified. AhR expression was transiently increased after MCAO, whereas TNFα expression was continuously enhanced. AhR is reported to be induced by NF-κB RelA as described above, although TNFα is upregulated activator protein 1 (AP-1) in addition to NF-κB and furthermore TNFα is reported to elicit NF-κB and AP-1 expression [33]. This positive feedback might be involved in sustained TNFα upregulation observed in this study.

Although AhR has been recognized as having a role in the metabolism of harmful exogenous compounds, endogenous ligands such as indole-3-carbinol and KYN have been discovered. Cuartero et al. reported that KYN levels are increased in the ischemic brain, probably because of upregulation of the expression of the KYN synthesizing enzyme tryptophan 2,3-dioxygenase [6]. We previously reported that the expression of another KYN-synthesizing enzyme, IDO, is upregulated downstream of AhR and the NF-κB subunit RelB [34]. IDO1 mRNA expression was increased in the ischemic region 3 and 6 h after MCAO in this study. Therefore, KYN can act as an endogenous ligand that might activate AhR and subsequently induce, at least in part, ischemic brain injury.

Vasogenic edema, which is induced by BBB rupture and is occasionally lethal because it can cause severe brain damage such as exencephaly, is characterized by extravasation of fluid and its extracellular accumulation in the cerebral parenchyma. Increased evidence has shown that cytokines released from activated microglia, such as TNFα and interleukin-6, can increase the permeability of endothelial cells followed by BBB disruption [35,36]. In this study, TNFα expression was induced in an AhR-dependent manner 3 h after MCAO, and vasogenic edema was detected 6 h after MCAO [8]. Therefore, TNFα might partially be responsible for edema formation. TBARS levels increased in the ischemic region of the brain, indicating that oxidative brain damage was induced by MCAO. ROS can increase cytokine expression by NF-κB activation; one possible mechanism is the inactivation of oxidation-sensitive phosphatases such as protein tyrosine phosphatases by oxidation of active site cysteine, which causes prolonged phosphorylation and activation of IκB kinase. IκB kinase phosphorylates IκB and enhances its degradation by the proteasome, and thus, NF-κB released from IκB translocates to the nucleus to activate transcription [37]. We previously reported that microglial ROS activate TNFα and IL-1β transcription [9], suggesting that activated microglia increase cytokine expression via oxidative stress during ischemia. Collectively, AhR can directly activate cytokine transcription and AhR increases NOX activity by increasing the expression of p47phox to elicit oxidative stress, which contributes to cytokine transcription. Both mechanisms might occur in the ischemic brain due to BBB rupture and subsequent vasogenic edema progression.

Phagocytic NOX is a multicomponent enzyme comprised of the membrane-bound flavocytochrome NOX2 associated with p22phox and several cytosolic proteins, including p40phox, p47phox, p67phox and rac1 [26]. The phosphorylation of p47phox is essential for the activation of this complex in intact cells, and activated NOX produces many superoxide anion radicals responsible for host defense. p47phox has been reported to be a target of AhR [27], and we showed that the two DREs in the promoter region of p47phox are functional. In addition, we demonstrated that superoxide produced by NOX is involved in proinflammatory cytokine transcription in THP-1 macrophages. It has been reported that p47phox can control inflammatory reactions via ROS production in macrophages [38]. Thus, the AhR-p47 axis might have an important role in oxidative stress and inflammation. Further study is needed to show the direct relationship between AhR-induced p47 and ischemic brain damage.

Four alleles encoding different forms of AhR are known to exist among different laboratory mouse strains. The Ah^b^ alleles, including Ah^b−1^, Ah^b−2^ and Ah^b−3^, encode a form of AhR that binds aromatic hydrocarbons with high affinity, while the Ah^d^ allele encodes a form of AHR that binds aromatic hydrocarbons with low affinity. The difference in the ligand-binding affinity of these forms of AhR is due to differences in the residue at position 375; a valine residue is present at position 375 in the Ah^d^ allele, whereas an alanine residue is found at this site in the three forms of the Ah^b^ allele [39]. ICR mice have an alanine residue at position 375, and thus, are considered to carry the Ah^b^ allele, which binds aromatic hydrocarbons with high affinity. On the other hand, AhR responsiveness has been reported to be determined by the expression levels of AhR and its binding partner ARNT [40]. ICR mice exhibit much lower expression levels of AhR than C57BL/6 mice [41], and thus, AhR shows low responsiveness to its ligands in these mice. Of note, in wild-type human AhR, position 381 (equivalent to position 375 in mice) is a valine residue; thus, wild-type human AhR has low affinity [42]. The extent of the AhR response in ICR mice is reportedly very similar to that of the AhR response in humans [41]. To compare the action of AhR between mice and humans, it might be important to determine AhR responsiveness by considering AhR expression and the particular AhR allele.

## 5. Conclusions

Microglial AhR can induce oxidative stress and inflammatory reactions during ischemia, which causes vasogenic edema progression and subsequent brain injury (Figure 12). AhR plays an important role in the upregulation of p47phox expression (Figure 12). AhR might be a pharmacological target for improving prognosis after ischemic stroke.

## Figures and Tables

**Figure 1 cells-10-00718-f001:**
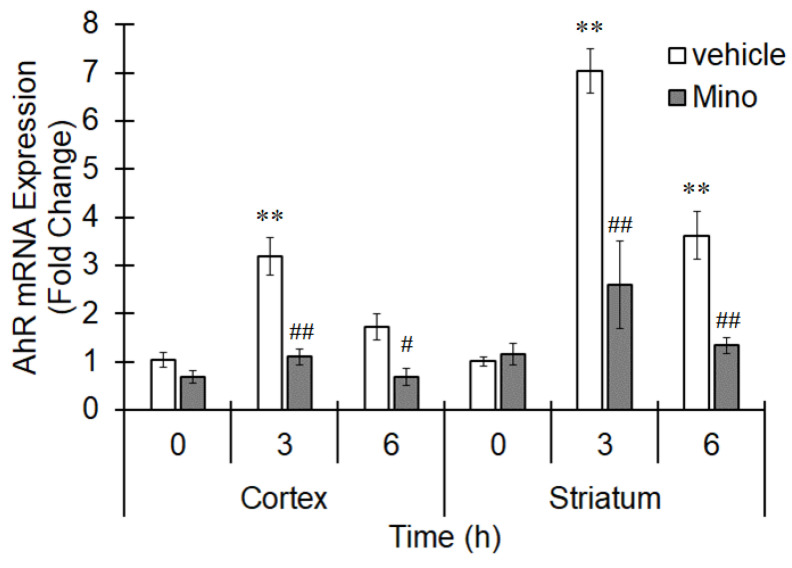
Increased AhR expression in microglia by middle cerebral artery occlusion (MCAO). Mice were intraperitoneally treated with minocycline (Mino, Wako Pure Chemical, Osaka, Japan; 100 mg/kg) 1 h before MCAO. The middle cerebral artery was occluded for 0 (sham operation), 3 or 6 h. The brain was excised, and the mRNA expression of AhR in the cortex and striatum was evaluated by real-time PCR. The values represent the mean ± S.E. (*n* = 5). The data were analyzed using one-way ANOVA followed by Dunnett’s test or Student’s *t*-test. ** *p* < 0.01 vs. the 0 h group. ^#^
*p* < 0.05 and ^##^
*p* < 0.01 vs. the vehicle group.

**Figure 2 cells-10-00718-f002:**
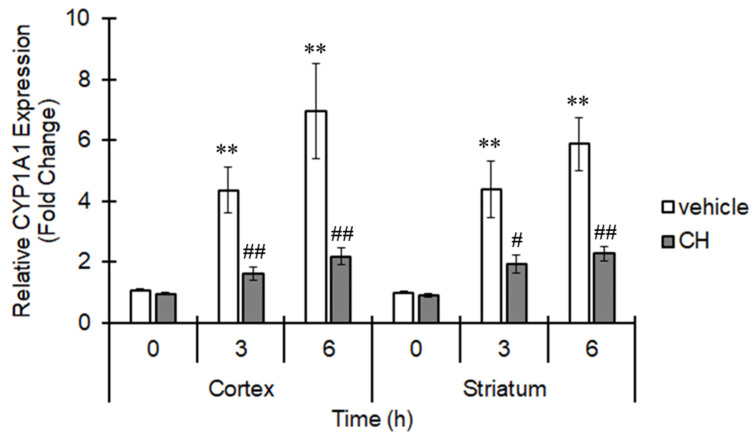
Increment of CYP1A1 expression by MCAO. Mice were intraperitoneally treated with CH223191 (CH, Sigma-Aldrich; 10 mg/kg) 1 h before MCAO. The middle cerebral artery was occluded for 0 (sham operation), 3 or 6 h. The brain was excised, and the mRNA expression of CYP1A1 in the cortex and striatum was evaluated by real-time PCR. The values represent the mean ± S.E. (*n* = 4). The data were analyzed using one-way ANOVA followed by Dunnett’s test or Student’s *t*-test. ** *p* < 0.01 vs. the 0 h group. ^#^
*p* < 0.05 and ^##^
*p* < 0.01 vs. the vehicle group.

**Figure 3 cells-10-00718-f003:**
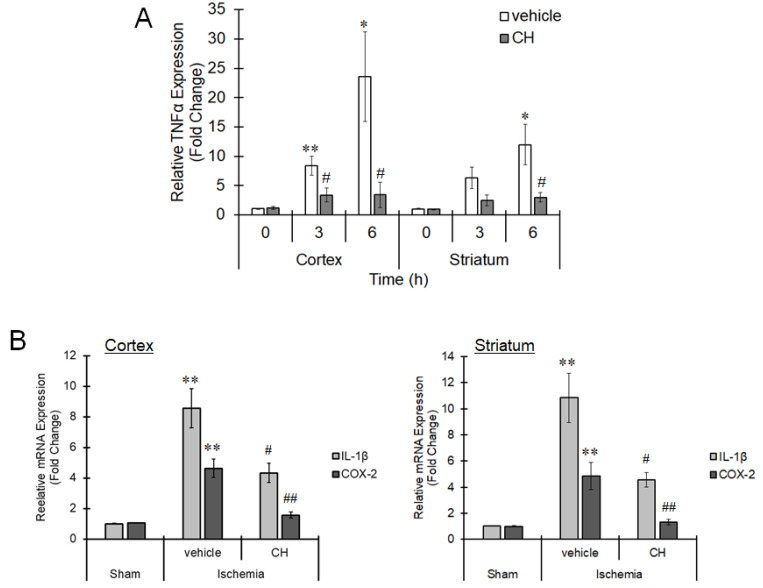
AhR-dependent proinflammatory cytokine upregulation in the ischemic brain. Mice were intraperitoneally treated with CH223191 (CH, 10 mg/kg) 1 h before MCAO. The middle cerebral artery was occluded for 6 h. The brain was excised, and the mRNA expression of (**A**) TNFα at 0, 3 and 6 h, (**B**) IL-1β and COX-2 at 3 h in the cortex and striatum was evaluated by real-time PCR. The values represent the mean ± S.E. (*n* = 5). The data were analyzed using one-way ANOVA followed by Dunnett’s test or Student’s *t*-test. (**A**) * *p* < 0.05 and ** *p* < 0.01 vs. the 0 h group. ^#^
*p* < 0.05 vs. the vehicle group. (**B**) ** *p* < 0.01 vs. the sham group. ^#^
*p* < 0.05 and ^##^
*p* < 0.01 vs. the vehicle group.

**Figure 4 cells-10-00718-f004:**
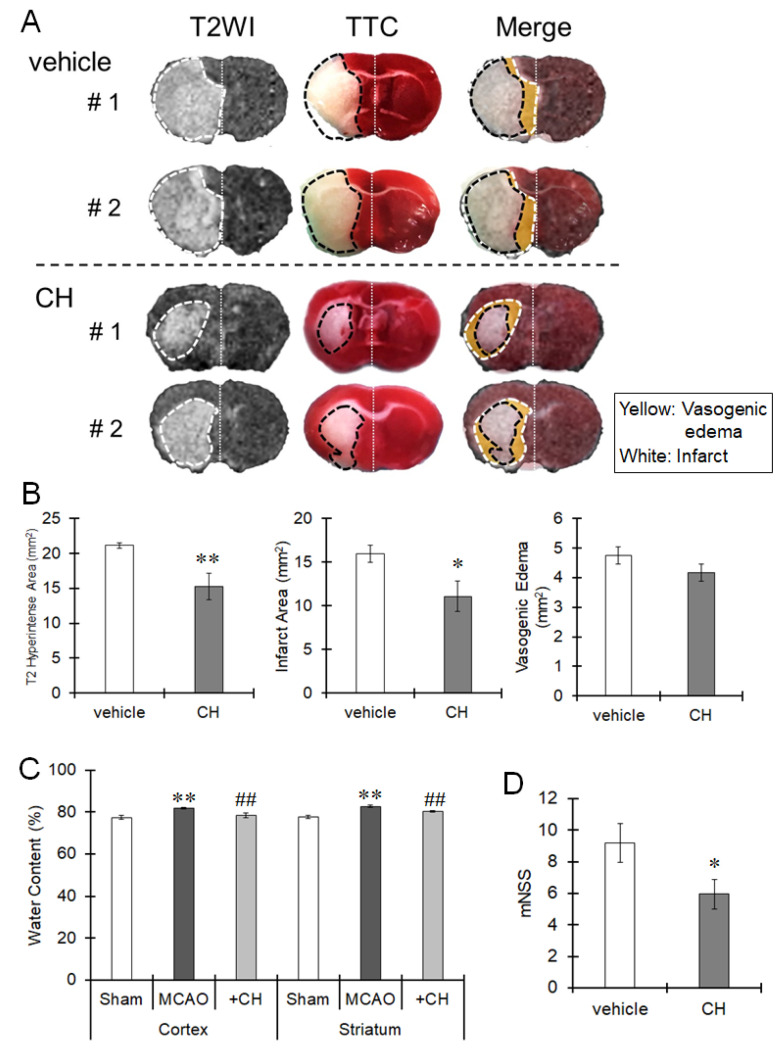
Protective effects of AhR on MCAO-induced brain damage. Mice were intraperitoneally treated with CH223191 (CH, 10 mg/kg) 1 h before MCAO. The middle cerebral artery was occluded for 24 h. (**A**) T2WIs and images of 2,3,5-triphenyltetrazolium chloride (TTC) staining were merged to identify the infarct, vasogenic edema region and normal regions. The yellow region in the merged images is the vasogenic edema region. (**B**) The T2 hyperintense lesion area, infarct (TTC-unstained) area and vasogenic edema area were quantified by ImageJ software. The values represent the mean ± S.E. (*n* = 8 animals in each group). Statistical analyses were performed using one-way ANOVA followed by Student’s *t*-test. * *p* < 0.05 and ** *p* < 0.01 vs. the vehicle group. (**C**) The brain was excised, and water content in the cortex and striatum was measured. The values represent the mean ± S.E. (*n* = 6 animals in each group). The data were analyzed using one-way ANOVA followed by Student’s *t*-test. ** *p* < 0.01 vs. the vehicle group. ^##^
*p* < 0.01 vs. the MCAO group. (**D**) Behaviors were monitored, and mNSSs were determined according to the criteria described in the Material and Methods section 24 h after MCAO. The values represent the mean ± S.E. (*n* = 9 animals in each group). The data were analyzed using one-way ANOVA followed by Student’s *t*-test. * *p* < 0.05 vs. the vehicle group.

**Figure 5 cells-10-00718-f005:**
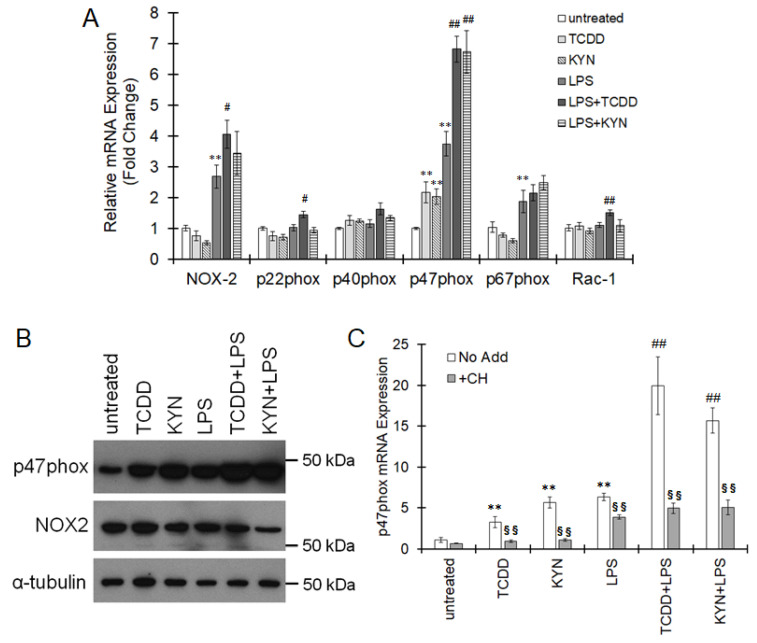
Increased expression of p47phox by AhR stimulation in human macrophages. (**A**,**B**) THP-1 macrophages were treated with 2,3,7,8-tetrachlorodibenzo-p-dioxin (TCDD) (10 nM) or kynurenine (KYN) (50 μM) for 6 h in the presence or absence of lipopolysaccharide (LPS) (10 ng/mL). (**A**) The expression of NADPH oxidase (NOX)-2, p22phox, p40phox, p47phox, p67phox and Rac-1 was assessed by real-time PCR. The values represent the mean ± S.E. (*n* = 4). The data were analyzed using one-way ANOVA followed by Dunnett’s test or Student’s *t*-test. ** *p* < 0.01 vs. the untreated group. ^#^
*p* < 0.05 and ^##^
*p* < 0.01 vs. the LPS-treated group. (**B**) The protein expression of p47phox and NOX2 was measured by immunoblotting. The results are representative of 3 independent experiments. (**C**) THP-1 macrophages were pretreated with the AhR antagonist CH223191 (10 μM) 20 min before treatment with TCDD (10 nM), KYN (50 μM) or LPS (10 ng/mL). Six hours after AhR agonist or LPS treatment, total RNA was extracted from the cells, and then p47phox mRNA levels were measured by real-time PCR. The values represent the mean ± S.E. (*n* = 4). The data were analyzed using one-way ANOVA followed by Dunnett’s test or *t*-test. ** *p* < 0.01 vs. the untreated group. ^##^
*p* < 0.01 vs. the LPS-treated group. ^§§^
*p* < 0.01 vs. the untreated group.

**Figure 6 cells-10-00718-f006:**
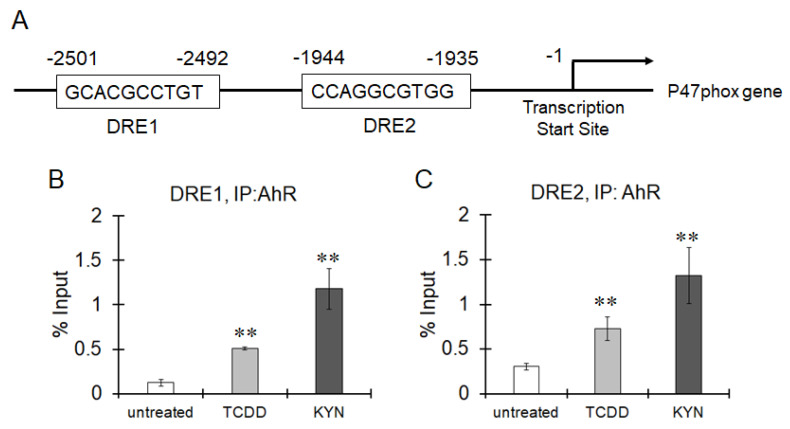
Recruitment of AhR to the DREs in the p47phox promoter region. (**A**) Structure of the human P47phox promoter. (**B**,**C**) THP-1 macrophages were treated with TCDD (10 nM) or KYN (50 μM) for 1 h. ChIP was performed to assess the binding of AhR to the dioxin response elements (DREs) within the human p47phox promoter region. Human genomic DNA (input for precipitation) was used as a positive control, and immunoprecipitation with a nonspecific antibody (IgG) was performed as a negative control. The resulting precipitants were analyzed by real-time PCR, and the percent input was calculated. The values represent the mean ± S.E. (*n* = 3). The data were analyzed using one-way ANOVA followed by Dunnett’s test. ** *p* < 0.01 vs. the untreated group.

**Figure 7 cells-10-00718-f007:**
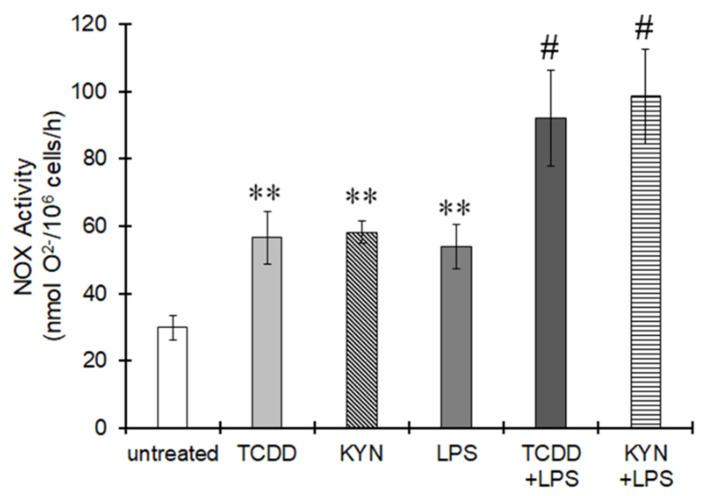
Increase in NOX activity by AhR ligands in macrophages THP-1 macrophages were treated with TCDD (10 nM) or KYN (50 μM) for 6 h in the presence or absence of LPS (10 ng/mL). NOX activity was measured by the cytochrome c method. The values represent the mean ± S.E. (*n* = 4). The data were analyzed using one-way ANOVA followed by Dunnett’s test or Student’s *t*-test. ** *p* < 0.01 vs. the untreated group. ^#^
*p* < 0.05 vs. the LPS-treated group.

**Figure 8 cells-10-00718-f008:**
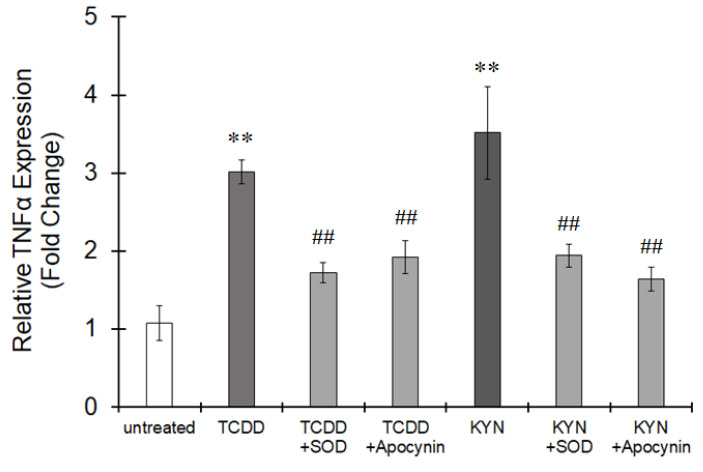
Upregulation of TNFα expression by ROS derived from NOX under AhR stimulation. THP-1 macrophages were treated with TCDD (10 nM) or KYN (50 μM) for 6 h in the presence or absence of superoxide dismutase-polyethylene glycol (SOD) (100 units/mL) or apocynin (300 μM). The expression of TNFα was assessed by real-time PCR. The values represent the mean ± S.E. (*n* = 5). The data were analyzed using one-way ANOVA followed by Dunnett’s test or Student’s *t*-test. ** *p* < 0.01 vs. the untreated group. ^##^
*p* < 0.01 vs. the TCDD- or KYN-treated group.

**Figure 9 cells-10-00718-f009:**
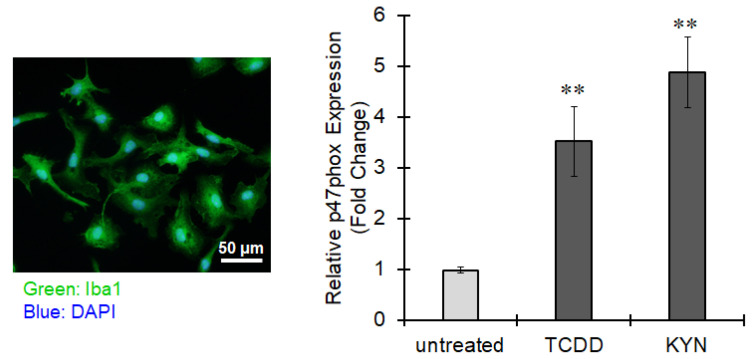
p47phox upregulation by AhR agonists in mouse primary microglia. (Left panel) Representative image of cultured mouse primary microglia stained by Iba1 (green) and DAPI (blue). (Right panel) Mouse primary microglia were treated with TCDD (10 nM) or KYN (50 μM) for 6 h. The expression of p47phox was assessed by real-time PCR. The values represent the mean ± S.E. (*n* = 4). The data were analyzed using one-way ANOVA followed by Dunnett’s test. ** *p* < 0.01 vs. the untreated group.

**Figure 10 cells-10-00718-f010:**
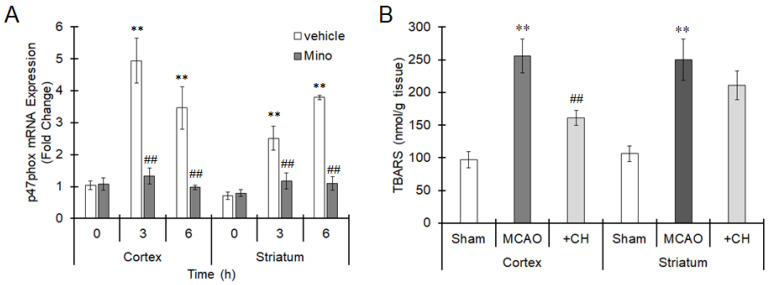
Increase in p47phox expression and oxidative stress under ischemic conditions. (**A**) Mice were intraperitoneally treated with minocycline (Mino, 100 mg/kg) 1 h before MCAO. The middle cerebral artery was occluded for 0 (sham operation), 3 or 6 h. The brain was excised, and mRNA expression of p47phox in the cortex and striatum was evaluated by real-time PCR. The values represent the mean ± S.E. (*n* = 5). The data were analyzed using one-way ANOVA followed by Dunnett’s test. ** *p* < 0.01 vs. the 0 h group. ^##^
*p* < 0.01 vs. the vehicle group. (**B**) Mice were intraperitoneally treated with CH223191 (10 mg/kg) for 1 h, and then the middle cerebral artery was occluded for 24 h. The brain was excised, and lipid peroxide levels in the cortex and striatum were measured. The values represent the mean ± S.E. (*n* = 5). The data were analyzed using one-way ANOVA followed by Student’s *t*-test. Multiple comparisons were made using Holm’s correction. ** *p* < 0.01 vs. the sham group. ^##^
*p* < 0.01 vs. the MCAO group.

**Figure 11 cells-10-00718-f011:**
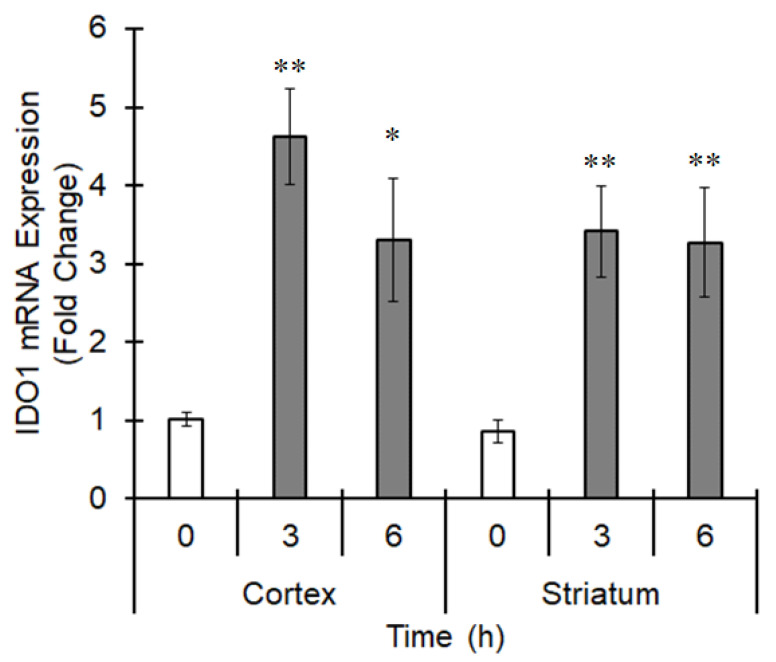
Upregulation of IDO1 expression during ischemia. The middle cerebral artery was occluded for 0 (sham operation), 3 or 6 h. The brain was excised, and the mRNA expression of IDO1 in the cortex and striatum was evaluated by real-time PCR. The values represent the mean ± S.E. (*n* = 5). The data were analyzed using one-way ANOVA followed by Dunnett’s test. * *p* < 0.05 and ** *p* < 0.01 vs. the 0 h group.

**Figure 12 cells-10-00718-f012:**
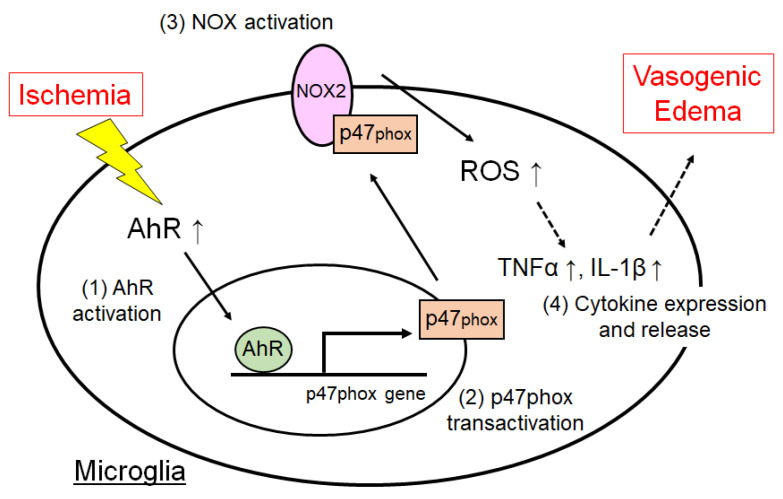
A possible mechanism underlying AhR-mediated inflammation and edema. (1) AhR expression is upregulated in microglia in the ischemic brain. (2) AhR can elicit a NOX subunit, p47phox expression. (3, 4) p47phox could increase cytokine expression via ROS generated from NOX and then induce neuroinflammation, followed by vasogenic edema.

**Table 1 cells-10-00718-t001:** Primers used for real-time PCR analysis.

Target	Forward Primer	Reverse Primer
Human NOX2	GGATGAGTCTCAGGCCAATCA	TCATTATCCCAGTTGGGCCG
Human p22phox	TACTATGTTCGGGCCGTCCT	GCACAGCCGCCAGTAGGTA
Human p40phox	GAGAGGTGAACTCAGCCTGG	TTCAAAGTCACTCTCGGCCC
Human p47phox	AGTACCGCGACAGACATCAC	CGCTCTCGCTCTTCTCTACG
Human p67phox	CTTGAACCAGTTGAGCTGCG	TTGTTTCTGGCCTGGTGACA
Human Rac-1	AAACCGGTGAATCTGGGCTT	AAGAACACATCTGTTTGCGGA
Human TNFα	TCCTTCAGACACCCTCAACC	AGGCCCCAGTTTGAATTCTT
Human β-actin	GGACTTCGAGCAAGAGATGG	AGCACTGTGTTGGCGTACAG
Mouse p47phox	CTGGAGGGCAGAGACAATCCA	CTGCTTCTCACACAGCGGA
Mouse AhR	ACCAGAACTGTGAGGGTTGG	TCTGAGGTGCCTGAACTCCT
Mouse CYP1A1	GGCCACTTTGACCCTTACAA	CAGGTAACGGAGGACAGGAA
Mouse TNFα	ATGGCCTCCCTCTCATCAGT	CTTGGTGGTTTGCTACGACG
Mouse IL-1β	AGCTTCCTTGTGCAAGTGTCT	GCAGCCCTTCATCTTTTGGG
Mouse COX-2	AGCCAGGCAGCAAATCCTT	CAGTCCGGGTACAGTCACAC
Mouse β-actin	AGCCATGTACGTAGCCATCC	CTCTCAGCTGTGGTGGTGAA

**Table 2 cells-10-00718-t002:** Primers used for the ChIP assay.

Target	Forward Primer	Reverse Primer
p47 DRE1	ATTAGCCGGACATGGTGGTG	ATGCAGTGGCATGATCTCGG
p47 DRE2	GCCAACAGGGTGATACCCCT	AGCTTCCCAAGTAGCTGGGA

## Data Availability

The data that support the findings of this study are available from the corresponding author (Y.I.) upon reasonable request.

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
