# Peer review of "Involvement of the Microglial Aryl Hydrocarbon Receptor in Neuroinflammation and Vasogenic Edema after Ischemic Stroke"

_cells, 2021, doi:10.3390/cells10040718_

Round 1

Reviewer 1 Report

COMMENTS TO THE AUTHOR (MANUSCRIPT cells-1129652)

In this manuscript, authors analyze the role of the aryl hydrocarbon receptor in neuroinflammation and vasogenic edema after ischemic stroke. The topic is interesting and adequate to the journal. A point that deserves a better explanation is the use of minocycline as inhibitor of microglial activation. Actually, this is the only link used in the paper to indicate that microglial AhR has a role in these processes. For example, the reference Kobayashi et al. (2013) Cell Death & Disease 4, e525 explain the mechanism of minocycline.

Minor changes:

- The legend of the figure 2 (second line) needs a rewriting.

- Lines 295-297 have to be deleted.

- The conclusion also needs a rewriting. The first sentence is incomplete.

Reviewer 2 Report

The article by Tanaka et al, entitled “Involvement of the microglial aryl hydrocarbon receptor in neuroinflammation and vasogenic edema after ischemic stroke”, deals with the role of AHR in the cell response after ischemia. They show that activation of the receptor induces a ROS production by NADPH oxidase which in turn leads to the expression of TNF-a. They finally propose AhR as a therapeutical target to reduce inflammatory reaction during ischemia.

The results are interesting and mainly support the conclusion. Nevertheless, before publication some points have to be clarified.

Line 43 (ref2) : specify that those results are found in microglia

Line 50 : what do the authors mean by “strong” agonist. The Kd of kynurenin for AhR is around uM whereas the Kd for dioxin is around nM.

Table 1: check the first couple of primers which blast NOX3 and not NOX2 gene.

As shown in fig 1 and 8, have the authors measured the production of TNFa after minocycline exposure ?

Line 205 : the authors write that the vasogenic edema region rarely spread into the cerebral cortex following pretreatment with CH223191. Does that mean that without CH223191, the vasogenic edema region does spread into the cerebral cortex ?

Line 209 : “mNSSs were worsened by ischemia ” … compared to what ?

Line 211: the authors conclude that AHR could promote the progression of vasogenic edema. It is not clear based on the description of the result from fig 3, how this conclusion is made.

Fig 3 : indicate on the figure what does refer exactly to infarct, vasogenic edema ..

The third panel of B is the surrounding vasogenic edema and not edema area.

Fig 4 : Could the author explain the discrepancies between expression of p47phox after LPS+TCDD exposure in fig A (6-7 fold) and in fig C (20 fold). The values after LPS or KYN exposures are more close.

The author should measure also AHR target genes (like CYP1A1) as positive control, to show the effect of CH on AHR pathway.

Fig 6 and fig 7 : why the author do not use CH223191 ?

Fig 7 : what is the effect of SOD or apocynin alone ?

Line 292 : p47phox expression is upregulated “through activation of” microglia.

Line 295/297 : this looks like internal comments of authors and not part of the article. To be removed.

Line 315 : even though the AhR gene is overexpressed, to conclude with confidence on the AHR involvement, protein expression should be analyzed and specific invalidation of gene by siARN or KO should be done.

Line 342 : it is not possible to add new result on the discussion part. If the author think this is important for their research, those figure/results must be move in the result part. The Supp fig 1 could be add in fig 8B.

Line 361 : can one really talk about oxidative stress ? It is more the normal function of NOX which products ROS as second messenger for cell response.

Line 381 : “whereas” instead of “which”.

Fig 9 : to be complete, and make a kind of graphical abstract, add SOD, Apocynin, CH, MCAO and Mynocyclin. Also; put p47 on the membrane where it should stand, and maybe AhR in the nucleus where it will have it’s transcriptional activity/

Reviewer 3 Report

Reviewer; The submitted manuscript suggests the role of aryl hydrocarbon receptor in neuroinflammation after ischemic stroke. Authors provided several results from various in vivo studies and macrophage results. I have a few concerns regarding necessary for the manuscript.

Major comments:

  1. AhR is expressed only in microglia? Although the authors pointed out that MCAO-induced inflammation was induced by AhR in microglia with minocycline or CH223191, it would be better to provide the evidence that the MCAO-induced effect was derived from AhR in microglia. Because astrocytes are enriched cells in brain and they mediate neuroinflammation in response to MCAO, it would be nice to provide the results or add references that expression of AhR in neuronal cells and AhR in microglia is involved in MCAO-induced inflammation. As THP-1 macrophages are not brain microglia, it would be better to provide the results from microglia.
  2. It seems gene expression pattern of AhR and TNFa is little bit inconsistent. AhR gene expressions was transiently increased after MCAO, whereas TNFa expression was continuously enhanced in same condition. Please describe the difference of gene expression between AhR and TNFa.
  3. In chip assay, positive control or raw data was not provided. Please have a description in detail or provide the data.
  4. In figure 6, NOX activity was enhanced by AhR ligands in THP-1 macrophage, whereas in figure 4B, NOX2 expression was not significantly changed in human macrophages. Please describe this inconsistence.
  5. In figure 3, CH suppresses both vasogenic edema and infarct area, then microglial AhR would have protective effect against MCAO-induced brain damage. Therefore, it would be better to change the title.

Minor comments:

  1. Please use the symbol, such as NF-kB.
  2. Correct the typographical errors, such as stratum, imput, and so on.
  3. Description of Figure 5C legend was absent.
  4. In submitted manuscript, English grammar should be carefully checked.

Reviewer 4 Report

This paper evaluates the AhR implication in neuroinflammation and vasogenic edema in a mouse MCAO model and THP-1 human macrophages. Authors report that MCAO upregulated AhR expression in microglia. The increased TNFα observed after MCAO and linked to AhR favored edema progression and worsened neurological scores. These effects were suppressed by the AhR antagonist, CH223191. In THP-1 human macrophages, NADPH oxidase (NOX) subunit p47phox significantly increased after AhR ligands, especially under LPS. Suppression of NOX activity or reduction of superoxide decreased the TNFα expression induced by AhR activation. In MCAO, P47phox expression was upregulated in microglia by ischemia. Authors conclude that AhR in microglia is involved in neuroinflammation and subsequent edema after MCAO via p47phox expression up- regulation and oxidative stress.

The main shortcomings of this work are: 1) authors claim “the involvement of microglia” without any evidence of an involvement of these cells in the reported effects. The only indirect evidence of the possible role of microglia is the effect of minocycline. In addition, data were obtained in mice and human monocytic cells and the latter experimental model was considered as microglia; 2) most of the data do not represent a novelty. Several studies addressed the role of AhR in neuroinflammation after ischemic stroke using different AhR antagonists and ligands and methods. In addition, also the relationship between AhR and TNF alpha and the P47phox role in inflammation via ROS production in macrophages has been previously reported (2021 Feb;18(2):259-268. doi: 10.1038/s41423-020-00585-5).

Specific points:

Title and abstract should be changed and better address the focus of study (most experiments address the role of p47phox in human monocytic cells).

Methods should be expanded (more information)

In the legend of the figures, when authors report data on mice they make a recurrent mistake, that is “…treated with... for 1 h”. I suppose authors means 1 h before.

Figure 9 appears an oversimplification. It should be deleted or modified unless authors demonstrate these effects on microglia and not monocytes.

The data concerning IL-1B and cyclooxygenase2 reported as “data not shown” in the discussion section could be reported in the manuscript.

About one-third of the references are from previous papers the same research group. It is suggested to refer also to studies of other authors.

Correct “stratum” with “striatum”

Round 2

Reviewer 3 Report

All issues have been addressed

Author Response

Thank you very much for the review.

Reviewer 4 Report

The manuscript has been improved

Author Response

Thank you very much for the review.